# Prime Editing for Human Gene Therapy: Where Are We Now?

**DOI:** 10.3390/cells12040536

**Published:** 2023-02-07

**Authors:** Kelly Godbout, Jacques P. Tremblay

**Affiliations:** 1Centre de Recherche du CHU de Québec-Université Laval, Quebec City, QC G1V 4G2, Canada; 2Department of Molecular Medicine, Faculty of Medicine, Laval University, Quebec City, QC G1V 0A6, Canada

**Keywords:** prime editing, gene therapy, inherited diseases, genetic diseases, CRISPR/Cas9

## Abstract

Gene therapy holds tremendous potential in the treatment of inherited diseases. Unlike traditional medicines, which only treat the symptoms, gene therapy has the potential to cure the disease by addressing the root of the problem: genetic mutations. The discovery of CRISPR/Cas9 in 2012 paved the way for the development of those therapies. Improvement of this system led to the recent development of an outstanding technology called prime editing. This system can introduce targeted insertions, deletions, and all 12 possible base-to-base conversions in the human genome. Since the first publication on prime editing in 2019, groups all around the world have worked on this promising technology to develop a treatment for genetic diseases. To date, prime editing has been attempted in preclinical studies for liver, eye, skin, muscular, and neurodegenerative hereditary diseases, in addition to cystic fibrosis, beta-thalassemia, X-linked severe combined immunodeficiency, and cancer. In this review, we portrayed where we are now on prime editing for human gene therapy and outlined the best strategies for correcting pathogenic mutations by prime editing.

## 1. Introduction

Gene therapy offers enormous potential in the treatment of genetic diseases. Its potency lies in addressing the genetic root of the problem, unlike traditional medicines, which only treat the symptoms. By correcting the mutations, gene therapy has the potential to cure hereditary diseases. Sherkow et al. [1] defined gene therapy as “the intentional, expected permanent, and specific alteration of the DNA sequence of the cellular genome, for a clinical purpose”. The first approved gene therapy occurred in 1990, when a foreign gene was inserted into a kid’s immune cells [2]. Gene therapies first took the form of DNA insertion into the host genome [3]. In the 2000s, tools allowing the introduction of modifications at specific target sites in the genome were developed, including zinc-finger nucleases (ZFNs) [4] and transcription activator-like effector nucleases (TALENs) [5].

A milestone in the development of gene therapies was the discovery of CRISPR/Cas9 in 2012. This system involves a Cas9 nuclease that induces a double-strand DNA break at a precise place in the genome. The Cas9 is directed at the right sequence in the genome by a guide RNA. This guide is a single RNA strand complementary to an 18–24 nucleotides (nt) sequence in the genome [6,7]. When the complex is fixed on the complementary DNA sequence and the Cas9 recognizes a protospacer adjacent motif (PAM), the complex is activated. This PAM sequence varies depending on the microorganism of origin of the Cas9. For example, the most widely used Cas9 is from *Streptococcus pyogenes* (SpCas9) and recognizes the PAM 5′-NGG-3′ [8]. Once this small sequence is recognized, Cas9 will cut 3 nt upstream of the PAM [9]. The cell will then repair its DNA by non-homologous end joining (NHEJ), microhomology-mediated end joining (MMEJ), or homology-directed repair (HDR). NHEJ is an imprecise mechanism where broken ends of DNA are joined together, which often leads to insertions or deletions of nucleotides [10]. MMEJ is also an imprecise mechanism that can lead to undesired insertions, deletions, or even translocations [11]. This mechanism works by aligning short homologous sequences that are between the broken ends [12,13]. HDR repairs the damage using a homologous donor DNA, leading to a precise repair of the cut [10].

The evolution and refinement of the CRISPR/Cas9 technology have driven the development of base editing. This system exists in three versions: cytosine base editors (CBEs) [14], adenine base editors (ABEs) [15], and C to G base editors (CGBEs) [16]. CBE can install C > T and G > A mutations, while ABE can induce A > G and T > C mutations [17], and CGBE can generate C-to-G transversions [16,18]. Those editors change all the intended base pairs in a precise window (for example, CBE switches all C•G base pairs located in the window to T•A base pairs). Compared to CRISPR/Cas9, the advantages of this technology are that base editing does not require a double-strand break (the system uses a modified D10A Cas9) and does not need an exogenous DNA template, which leads to the reduction of unwanted indels. Base editing also leads to much more precise correction of the mutation by its ability to target a particular codon. However, those systems [19] cannot be applied when the change of a base pair in the window (other than the desired one) would lead to a non-silent mutation.

In October 2019, David R. Liu’s group released an outstanding discovery called prime editing [20]. This system can mediate targeted insertions, deletions, and all 12 possible base-to-base conversions. This mechanism makes DNA modifications with unprecedented precision and has substantial advantages over the traditional CRISPR/Cas9 and base editing systems (Table 1). Derived from the CRISPR/Cas9 system, this new technology is composed of a Cas9 nuclease fused with a reverse transcriptase (RT) at its 3′ extremity and a prime editing guide RNA (pegRNA) (Figure 1). The combination of Cas9 and RT forms the prime editor (PE). The pegRNA is composed of a spacer sequence, a primer binding site (PBS), a reverse transcriptase template (RTT), and a common region that binds to the Cas9 and the RT [19]. First, the complex binds to the DNA, guided by the spacer sequence in the pegRNA. The Cas9 recognizes a PAM and cuts 3 nt upstream. However, instead of creating a double-stranded cut, the modified H840A Cas9 from the prime editing induces a single-stranded nick [20]. Then, the PBS hybridizes to its complementary sequence located on the cut strand. Then, the RT will use the RTT as a template to transcribe the cut strand. At this point, one of the strands has a duplicated section, so the cell will have to remove one of the two sections to put back the DNA double-stranded again. The mismatch will be resolved by a 5′ flap or a 3′ flap. If a 3′ flap happens, the correction will be kept, but the editing will be lost if a 5′ flap occurs [20].

To introduce any modifications, the desired correction needs to be introduced in the RTT sequence, since it serves as a template for the transcription. There are currently several versions of the prime editor, the most popular being PE2 and PE3. PE3 is similar to PE2 but has an additional guide RNA that will induce a nick on the strand not initially cut by Cas9. It will promote the replacement of the unedited strand by forcing the cell to use the edited strand as a template. This increases the chances of retaining the edit at the mismatch repair step. When a position is given, this one is defined according to the reference system based on the initial Cas9 cleavage site. Thus, nucleotides downstream of this site will have a positive position (e.g., +5 means five nucleotides after the cut site towards 3′), and nucleotides upstream of the cut site will have a negative position (e.g., −5 means five nucleotides before the cut site towards 5′).
cells-12-00536-t001_Table 1Table 1Comparison of advantages and disadvantages of CRISPR/Cas9, base editing, and prime editing systems.
CRISPR/Cas9Base EditingPrime EditingOff-targeteffectsSignificant off-target effectsLittle or no off-target effectsLittle or no off-target effectsPossibility of non-specific indels at DSB site [21];DNA donor template can lead to plasmid integration in the genome;Possible genome-wide off-targets.No DSB;Bystander base edits within a narrow window of 4–10 nt [22];Genome-wide off-targets studies need to be made.No DSB;No bystander edits;Genome-wide off-targets studies need to be made.FlexibilityCan introduce insertions, deletions, and all types of substitutions.Can introduce C > T, G > A, A > G, T > C and C > G substitutions only.The consideration of bystander edits makes base editing more stringent on the possible sites [22].Can introduce insertions, deletions, and all types of substitutions.Less stringent PAM requirements [23].Programmability ^1^Only if a DNA donor template is givenYesYesEfficient in vivodeliveryCurrently possibleCurrently possible(but more difficult than CRISPR/Cas9 because of its larger size)Need to be improved(too big for conventional vehicles)^1^ Possibility to determine the issue of editing.


As prime editing can introduce insertions, deletions, and all types of substitutions, it has the potential to correct every mutation that causes a hereditary disease. Prime editing can also generate cell lines or animal models for specific mutations, driving forward research on several genetic diseases. This review summarizes what has been accomplished in preclinical studies using prime editing for human gene therapy. To date, prime editing has been attempted on liver, eye, skin, muscular, and neurodegenerative hereditary diseases, in addition to cystic fibrosis, beta-thalassemia, X-linked severe combined immunodeficiency, and cancer. We also discuss the best strategies used in those studies to guide future experiments on correcting pathogenic mutations by prime editing.

## 2. Application of Prime Editing to Liver Hereditary Diseases

There are several published studies on the correction by prime editing of mutations causing liver diseases or on the generation of animal models with these mutations (Table 2). Liver diseases are indeed currently the most studied field for human gene therapy by prime editing.

Schene et al. [24] were the first to publish an original article in 2020 about prime editing for liver diseases. They generated disease model organoids by prime editing and aimed to correct patient-derived disease models by prime editing. First, the authors introduced a 6 nt deletion in the *CTNNB1* gene that may lead to liver cancer. Using the PE3 system and a pegRNA composed of a 20 nucleotides spacer, a 12 nucleotides PBS, and a 17 nucleotides RTT, they introduced in liver organoids the deletion at the +1 position in 30% of the cells. By using PE3 and a silent mutation in the PAM (to avoid a second prime editing event on an already edited allele), they introduced the A > G substitution (at +7 position in the pegRNA) in 20% of the liver organoid clones in the *ABCB11* gene that causes bile salt export pump deficiency. Next, the authors tried to correct mutations in patient-derived intestinal cells and liver organoids. Using PE3, they successfully corrected 21% of the S210del in the *DGAT1* gene. However, they did not obtain any editing for the correction of the R1153H mutation in the *ABCB11* gene and the E342K mutation in the *SERPINA1* gene in patient-derived liver organoids. They used the PE3 system, but the authors mentioned that they did not conduct any pegRNA design optimization, which is crucial for good prime editing efficiency.

Liu et al. [25] also published on prime editing for liver disease. They worked on the correction of the E342K mutation in the *SERPINA1* gene which causes Alpha-1 antitrypsin (A1AT) deficiency. The E342K mutation is the most common one causing this disease. It is the same mutation that Schene et al. [24] failed to correct, highlighting that pegRNA design optimization is crucial. Liu et al. [25] first used PE2 to introduce the mutation into HEK293T cells. The classic PE2 requires an NGG PAM. Unfortunately, there were none near the site to be edited. Thus, they selected one that was more distant. Their pegRNA had a 20 nt spacer, a 13 nt PBS, and a 27 nt RTT. This pegRNA and PE2 achieved only 1.9% editing in HEK293T cells. However, with the same pegRNA and PE3, they achieved 9.9% editing. With this pegRNA and PE* [25] (PE that has a nuclear localization signal optimization), they obtained 6.4% of editing with PE2* and 15.8% with PE3*. They then attempted to correct in vivo a mutation in PiZ mice [25] (mouse model for the mutation E342K in the *SERPINA1* gene). To do so, they delivered prime editing’s plasmid DNA by hydrodynamic tail vein injection. They obtained 2.1% editing with PE2 and 6.7% with PE2*. AAVs capacity being at most 5 kb [32], prime editing cannot be delivered using a single AAV. To address that issue, the authors tested to deliver the split-intein prime editors via AAVs by tail vein injection. By injecting a low dose of dual AAV8-PE3 (2 × 10^11^ viral genome), they detected 0.6% of editing after two weeks, 2.3% after six weeks, and 3.1% after ten weeks. However, the authors hypothesized that because the PAM is far from the edited site, the efficiency of prime editing is not at its best and that the utilization of a nearer PAM will lead to greater efficiency.

In the same month that Liu et al. [25] released their study, Habib et al. [26] published an article on prime editing in hiPSCs (human induced pluripotent stem cells) by generating a doxycycline-inducible prime editing platform. They also attempted to correct by prime editing the E342K mutation (1024 G > A) in the *SERPINA1* gene in hiPSCs derived from a patient with alpha 1-antitrypsin (A1AT) deficiency. First, with their doxycycline-inducible prime editing platform using PE3, the authors demonstrated that creating hiPSC cell lines containing a desired mutation for the HEK3 and RNF2 locus was possible. They obtained similar conclusions by comparing their pegRNA constructs for these loci with the results in hiPSCs of David Liu’s group (the creators of prime editing) in HEK293T cells. For example, both groups concluded that, compared to the HEK3 locus, efficient prime editing at the RNF2 site required a longer PBS sequence. That conclusion suggests that the optimum PBS length depends on the sequence and, thus, might have the same optimal length even when used in different cell types. Moreover, transition substitutions (position +1 to +33), transversion substitutions (position +1 to +33), small indels (1–3 bp at +10 position), large deletions (up to 80 bp) and large insertions (up to 42 bp) are possible at the HEK3 target site in hPSCs. However, Habib et al. [26] failed to introduce small indels at the +17 and +21 positions for a mutation in the *RNF2* gene.

Another exciting aspect of Habib et al.’s [26] study is the section explaining why the PE3 system generates indel mutations. To understand what the main factor was, the authors tested five different scenarios: a single DSB induced by wild-type (wt) Cas9 and a sgRNA, a single nick induced by nCas9-RT and a sgRNA, a double DSB induced by wt Cas9 and two sgRNAs, a double nick induced by nCas9-RT and two sgRNAs, and finally the PE3 system (double nick induced by nCas9-RT, one pegRNA, and one sgRNA). From this experiment, the authors concluded that the unintended indels are not directly caused by double nicking but by the combinatory activity of the RT and the pegRNA.

The last section of this paper is about the correction by prime editing of the E342K mutation in the *SERPINA1* gene in hiPSCs derived from a patient with A1AT deficiency. The results only revealed a very low editing (0.83%), which was comparable to the editing made by ABE. Low editing efficiency might be caused by the low transfection efficiency.

In November 2021, Genesis Lung released his thesis [27] on the correction by prime editing of the E342K mutation in the *SERPINA1* gene that causes the A1AT deficiency. Lung worked on the same mutation as Schene et al. [24], Liu et al. [25], and Habib et al. [26], whose work has been described above. The last three mentioned authors failed to demonstrate a detectable correction of the mutation [25]. They used a PE that required an NGG PAM, which caused the targeted nucleotide to be far from the cut site, thus hindering good efficiency. In their original paper, Anzalone et al. [20] proposed an RTT length between 10 and 16 nt, which is much shorter than what Liu et al. [25] used (27 nt) [26]. Lung [27] attempted to correct the E342K mutation in the *SERPINA1* gene using a prime editor using another PAM. He first designed a variant of the prime editor that could use different PAMs (NGG, NGA, NGC) and conjugated them with pegRNAs that had different PBS (13 or 17 nt) and RTT lengths (between 10 and 20 nt). Liu reached up to 3% editing for the variants using an NGA PAM in HEK293T cells containing a lentivirus cassette with the E342K mutation. The best pegRNA had a spacer of 20 nt, a PBS of 13 nt, and an RTT of 20 nt. With PE3, he reached 3 to 5% editing. Those experiments were conducted by transfecting plasmids DNA with Lipofectamine 2000. He also tried to transfect the prime editing components in their RNA form, which can improve efficiency [33]. However, in this case, the efficiency was not significantly different between the use of DNA or RNA. He next tried to correct the E342K mutation in human primary fibroblasts. By using NGA-PE2, he obtained 1.99% of editing. In his thesis, Lung also described that overexpression of the three prime repair exonuclease 2 (*TREX2*) gene might decrease prime editing efficiency. It has been shown that this nuclear protein has 3′–5′ exonuclease activities [34]. Non-engineered pegs being really sensible to degradation by exonuclease, the presence of TREX2 may decrease prime editing efficiency by degrading pegs from their 3′ extremity, thus reducing the available number of pegs. Hence, the knockout of this gene may help prime editing. It is to note that since Lung’s master’s thesis is not a peer-reviewed article. Therefore, those results should be taken with caution.

Böck et al. [28]. worked on the editing of two other loci, dnmt1 and Pah^enu2^, both of which code for proteins expressed in the liver. They first tested two variations of the prime editor, the intein-split PE2∆RnH [28] and its unsplit version. AAV8 delivered the split version, and the unsplit version, being larger, was delivered by human adenoviral vector 5 (AdV). The latter virus can contain a larger cargo but is much more immunogenic. With the split version, they achieved 15% editing in vivo of the dnmt1 locus. Since they assumed that the unsplit version would give a better result, the authors also tested it in vivo, and they obtained much better results when the prime editor was not split. In neonatal mice, they obtained 58.2% of editing; in adult mice, they obtained 35.9%. Next, they tested the correction of a mutation at the Pah^enu2^ locus (F263S, c.835T > C), this mutation leading to phenylketonuria, a liver disease. They first tested their pegRNAs in an HEK293T cell line that had stably integrated exon 7 of the Pah^enu2^ allele. Their two best constructs had a spacer of 20 nt, a PBS of 13 nt, and an RTT of 16 or 19 nt. These two pegRNAs resulted in an editing of nearly 20%. The one with a 19 nt RTT had fewer off-targets; this pegRNA was thus chosen for the in vivo experiments. For the in vivo experiments, a mouse model of phenylketonuria (Pah^enu2^ mouse model [28]) was used. First, with the split version of the prime editor delivered by AAV8, they obtained meager results (less than 1% with PE2 and less than 2% with PE3). They then used the unsplit version of the prime editor delivered by an AdV. With the PE2 version in adult mice, they obtained only 2% editing. However, the results were better when they treated neonate mice. With PE2, they obtained 6.9% editing, and with PE3, they obtained 11.1% editing. This percentage was sufficient to lead to a therapeutic reduction of blood phenylalanine, without even inducing detectable off-target mutations and without leading to prolonged liver inflammation. Their results are encouraging. This project’s major problem in pursuing clinical translation is that they deliver a massive dose of virus (7 × 10^14^ vector genome/kg). In addition to being very expensive, using that amount of virus results in the induction of the immune system. They however noted that the percentage of edited hepatocytes was maintained even 12 weeks after the injection.

Jiang et al. [29] optimized prime editing to delete and replace long genomic sequences. The authors combined the PE with two pegRNAs, and they called it the PE-Cas9-based deletion and repair (PEDAR) method. They tested their method to remove a 1.38 kb pathogenic insertion in the *FAH* gene in a mouse model of tyrosinemia and replace it with a 19 bp sequence. They successfully delete the fragment and repair the deletion junction to restore FAH expression in the liver. They detected FAH-expressing hepatocytes on PEDAR-treated liver sections and obtained 0.76% of correction. Even if this is low, edited hepatocytes gained a growth advantage and eventually repopulated the liver. Indeed, forty days after the treatment, widespread FAH patches were observed in PEDAR-treated mouse liver sections and edited hepatocytes showed normal morphology.

Kim et al. [30] also worked on hereditary tyrosinemia type 1. They studied the correction by prime editing of a G > A point mutation (position +10 in their pegRNA) in the *FAH* gene in chemically derived hepatic progenitors (CdHs) from a mouse model of hereditary tyrosinemia (HT1 mice) [30]. After the treatment by prime editing, they grafted those cells into the liver of HT1 mice to study the repopulation of the liver by the corrected cells. First, the authors generated CdHs from HT1 mice. Next, they electroporated the cells with PE3, a sgRNA nicking at position -4, and a pegRNA that contained a spacer of 20 nt, a PBS of 11 nt, and an RTT of 15 nt. They obtained 2.3% editing without any off-target effects. Next, these authors tested the possibility of an ex vivo therapeutic transplantation of a corrected HT1-mCdHs-PE3b (chemically derived hepatic progenitors that are from HT1 mice and that have been treated with PE3b) cell population into the livers of HT1 mice. Because the bulk population of cells had a sufficient editing efficiency, they did not need to isolate cell clones. They thus directly grafted the bulk population of treated cells in the liver of HT1 mice. HT1-mCdHs-PE3b transplanted mice survived for more than 160 days, compared to control mice injected with PBS that died before 90 days. Liver damage of transplanted mice had also significantly decreased. After 140 days, the authors showed that the FAH-positive cell population in the HT1-mCdHs-PE3b transplanted liver repopulated the liver. Because of the repopulation of those cells, the percentage of edits in the liver increased from 2.3 to 34.3%.

Jang et al. [31] also studied the correction by prime editing of a mutation in the *FAH* gene. They worked on the G > A point mutation at the last nucleotide of exon 8. This mutation causes exon 8 skipping and results in loss of function of *FAH*, which causes hereditary tyrosinemia type 1. They first tested some pegRNAs in mutant Fah target sequence–containing HEK293T cells. The best one had a spacer of 20 nt, and the edit was at position +10 from the cut site. By using PE2, they obtained 18.7% of editing in vitro. They next tested their pegRNA in vivo in a mouse model of hereditary tyrosinemia type 1 (Fah^mut/mut^). The authors tested both PE2 and PE3. The first good news was that treated mice survived till the end of the experimental period (40 days for experiments with PE3 and 60 days for experiments with PE2), unlike the sick control mice which showed substantial weight loss and died before day 30 of the experiment. With these results, they demonstrated that the treatment prevented the mice from losing weight and prolonged their survival. Quantitative analyses were conducted following the mouse sacrifice. In the PE3-treated Fah^mut/mut^ mice, they observed that 12% of the Fah mRNA contained the exon 8, compared to the control Fah^mut/mut^ mice that did not have any. The frequency of FAH+ cells in the liver was also quantified. At day 40, PE3-treated mice had on average 61% FAH+ cells. At day 60, PE2-treated mice had on average 33% FAH+ cells. They also determined the percentage of editing by deep sequencing. They observed 11.5% editing in PE3-treated mice and 6.9% in PE2 treated-mice. This percentage was lower than the frequency of FAH+ cells because most hepatocytes are polyploid [35] and because hepatocyte DNA was mixed with nonparenchymal cell DNA. Therefore, the frequency of FAH+ cells is more indicative of clinical improvement. The authors also investigated the presence of indels at or near the targeted nucleotide and at potential off-target sites of the pegRNA. No off-target mutations were detected.

In short, for liver diseases, prime editing has been used to correct mutations causing DGAT1-deficiency, bile salt export pump deficiency, alpha-1-antitrypsin deficiency, phenylketonuria, and tyrosinemia type 1. Prime editing has also been used to generate cell or animal models for liver cancer, bile salt export pump deficiency, alpha-1-antitrypsin deficiency, and a liver disease caused by a mutation in the *DNMT1* gene.

## 3. Application of Prime Editing to Eye Hereditary Diseases

The development of a prime editing treatment for eye hereditary diseases may reach a clinical application faster because of the limited local delivery of the therapeutic agents. The eye has been the first human organ to receive a CRISPR/Cas9 treatment in clinical trials [36]. It is therefore interesting to review the preclinical studies of prime editing to correct eye hereditary diseases (Table 3).

In addition to work on prime editing for an *FAH* mutation, Jang et al. [31] studied prime editing for the R44X mutation causing Leber congenital amaurosis (LCA). This mutation is a C > T transition in exon 3 of the *RPE65* gene and leads to retinal degeneration and severe early-onset visual deterioration [39,40]. Jang et al. [31] first tested some pegRNAs targeting the R44X mutation in HEK293T cells. The best pegRNA had a spacer of 19 nt, a PBS of 9 nt, and an RTT of 14 nt. By using PE2, they obtained 14% of editing in vitro. They next tested their pegRNAs in vivo in *rd12* mice, a mouse model of RPE65-related LCA. They conducted subretinal injections of trans-splicing AAV (serotype 2) coding for PE2 and for a pegRNA. They obtained 6.4% editing. Considering that only 23% of the retinal pigment epithelium had received the AAV, the authors estimated at 28% the editing efficiency in regions that were exposed to prime editing components. The LCA mice treated by prime editing also had an improvement in their visual function. They demonstrated that the editing was highly precise, as no unintended edits, substitutions, or indels were detected near the mutation site or at the genomic DNA level.

Lin et al. [37] studied the generation by prime editing of a mouse model with a cataract disorder. They first tested the prime editing on mouse neuro-2a (N2a) cells. They used PE3 with a pegRNA that deletes 1 bp (a G) in exon 3 of the crygc gene. This mutation leads to a truncated gamma-crystallin protein, which causes nuclear cataracts in homozygous and heterozygous mutant mice. For the pegRNA, they used a spacer of 20 nt, a PBS of 13 nt, and an RTT of 10 nt. The additional sgRNA nicked 25 bp downstream the cut site. The authors succeeded in introducing this deletion in 80% of the cells. They next tried to generate a mouse model with a cataract disorder caused by the deletion of the G nucleotide in exon 3 of the crygc gene. Microinjections of the prime editing plasmids (the same as the one used in the first experiment) have been made in mouse embryos. They generated a mouse with a high G-deletion rate (38.2%) that displayed a nuclear cataract phenotype. They also showed that this mutation can be transmitted to future generations and that the progenies had a phenotypic inheritance of cataracts. The authors also checked for off-target indels and mutations but did not detect any. Lin et al. [37,38] also tried to repair this deletion in an N2a cell line containing the G-deletion. By using PE3, they succeeded in correcting this deletion in 33% of the cells.

Lv et al. [38] worked on the correction by prime editing of the c. 2234_2237del in the RPGR ORF15 that causes X-linked retinitis pigmentosa (XLRP), a disease that leads to blindness. The authors used an enhanced prime editor where a 20 nt Csy4 recognition site was fused with the pegRNA. This improvement reduced the complementary pairing between PBS at the 3ʹ end of pegRNA and part of the spacer at the 5ʹ end of pegRNA. For the pegRNA, they used a spacer of 20 nt, a PBS of 14 nt, and an RTT of 14 nt. They treated HEK293-mRO cells (cells that have c. 2234_2237del) with those constructions and obtained 12.05% editing.

## 4. Application of Prime Editing to Skin Disease

Hong et al. [41] worked on the correction by prime editing of two mutations (c.3631C > T and c.2005C > T) in the *COL7A1* gene causing recessive dystrophic epidermolysis bullosa (RDEB). Those mutations cause loss of function of the type VII collagen (C7) protein and lead to skin fragility. Prime editing was made ex vivo on patient-derived fibroblasts, and they were next transplanted into immunodeficient mice. The authors used PE3 and a pegRNA containing a 22 nt spacer, a 13 nt PBS, and a 14 nt RTT for prime editing (Table 4). In addition to correcting the mutation, the pegRNAs also introduced a silent mutation into the PAM. For the first mutation (c.3631C > T), editing was conducted 10 nt downstream from the nick site, and the sgRNA nicked 60 nt downstream. For the second mutation (c.2005C > T), the editing was conducted 12 nt downstream from the nick site, and the sgRNA nicked 56 nt upstream. The authors successfully corrected 10.5% of cells containing the first mutation and 5.2% of cells containing the second mutation. In vitro, the expression level of the C7 protein reached 55% in cell lysates and 37% in supernatant culture. The fibroblast adhesion properties and proliferation ability (which are decreased by the loss of function of this protein) were also improved in the edited fibroblasts. As the correction of the first mutation was the best, the in vivo transplantation step was only conducted with these cells. The restoration of the protein function could therefore be studied in vivo. First, they looked at whether the protein would localize correctly, i.e., at the dermal-epidermal junction (DEJ), after intradermal injection. Two weeks after injection, they saw that the human C7 protein was functional and localized linearly at the DEJ.

Subsequently, the authors reconstituted human skin grafts of patient-derived keratinocytes and fibroblasts on the back of immunodeficient mice. This experiment demonstrated that the gene corrected skin restored C7 protein deposition and anchored fibril formation at DEJ. It was shown that a C7 protein expression level of 35% is sufficient to provide mechanical stability of the skin in a DEB hypomorphic murine model [43]. Hong et al. [41] restored up to 49% of C7 protein expression, which means that the treatment could be effective enough to provide therapeutic improvement. The authors also observed that the protein was correctly deposited along the DEJ and that the edited fibroblasts showed enhanced proliferation compared with non-edited cells. This last point may explain why the levels of C7 restoration are higher than the percentage expected from the editing frequency at the genomic DNA level.

Petri et al. [42] studied the introduction by prime editing of the P301L mutation in the *TYR* gene (P302L for zebrafish). This mutation is responsible for oculocutaneous albinism and cannot be introduced in cells using base editing. In this study, the PE3 strategy was used, and prime editing components were delivered as ribonucleoproteins (RNP). They used a 4:1 pegRNA:PE protein ratio and a 10:1 molar ratio of pegRNA/sgRNA. They used a pegRNA with a spacer of 20 nt, a PBS of 10 nt, and an RTT of 15 nt. The edit site was at the +3 position, and the sgRNA used for PE3 cut at the position −83. With this combination, they obtained 8% editing. However, Petri et al. [42] observed significant unintended indels. The observed deletions, insertions, substitutions, potential RTT duplications, and potential pegRNA scaffold incorporations in some embryos.

## 5. Application of Prime Editing to Skeletal and Cardiac Muscle Diseases

The potential of prime editing has been tried on two muscular diseases: Duchenne muscular dystrophy, caused by a mutation in the *DMD* gene, and spinal muscular atrophy, caused by a mutation in the *SMN2* gene (Table 5).

Chemello et al. [44] worked on gene therapy for mutations in the *DMD* gene with the aim of inducing exon skipping to restore the reading frame and, thus, dystrophin expression. They tested in vitro a pegRNA possessing a 20 nt spacer, a 13 nt PBS, and a 15 nt RTT, with PE3 and a sgRNA cutting 52 nt downstream. In iPSC ∆Ex51 cells, they achieved 54% editing. They then differentiated these cells into cardiomyocytes to test the effect on dystrophin recovery. The edited cells expressed 39.7% of dystrophin compared to the healthy control. As the diseased cells showed arrhythmic defect problems, the authors tested the function of the treated cells compared to the diseased and healthy cells. The cells showed a percentage of arrhythmic calcium traces comparable to that of healthy cells.

Mbakam et al. [45,46] worked on the correction by prime editing of different mutations in the *DMD* gene that cause Duchenne muscular dystrophy. In their first article, Mbakam et al. [45] worked on inserting specific point mutations in exons 9, 20, 35, 43, 55, and 61 of the *DMD* gene. They first introduced the following mutations using PE2 in HEK293T cells: C > T in exon 9, C > T in exon 35, G > T in exon 20, C > T in exon 43, A > T in exon 55, and C > T in exon 61. The pegRNAs for exons 9, 35, and 55 produced better results using a PBS of 13 nt and an RTT of 13 nt. They, respectively, obtained 4, 6, and 3.5% of editing, and the mutations were, respectively, at position +3, +1, and +3 from the cut site. The pegRNAs for exons 20 and 43 produced better results using a PBS of 13 nt and an RTT of 10 nt. The mutations were, respectively, at position +10 and +1 from the cut site and the authors obtained 5 and 3.5%, respectively, for exons 20 and 43. Finally, for exon 61, the best pegRNA had a PBS of 15 nt and an RTT of 13 nt, and the edit site was at position +5 from the nick. This formulation gave 6% of editing. The authors next tested the editing in exon 35 with the same pegRNA but using the PE3 strategy with a sgRNA targeted at +57 nucleotides from the first nick site. It resulted in 20% editing of the desired mutation. Moreover, with the same pegRNA targeting with exon 35, they tried to mutate the PAM to avoid a second prime editing event on an already edited allele. This strategy led to 14% of editing using PE2 and 38% of editing using PE3. After, the authors studied the correction by prime editing of the c.428 G > A mutation in exon 61 of the *DMD* gene. For that correction, they used PE3, with a sgRNA targeted at +60 nt from the first nick site and a pegRNA having a spacing of 20 nt, a PBS of 14 nt, and an RTT of 16 nt. The edit site was at position +4 from the cut site. They electroporated the prime editing components in human myoblasts from a DMD patient with the c.428 G > A mutation. They obtained 8% of editing after one treatment and 29% after four treatments.

In their most recent article, Mbakam et al. [46] worked on the c.8713C > T mutation, located in exon 59 of the *DMD* gene. They first tried to introduce this mutation in HEK293T cells. By using PE2 and a pegRNA with a spacer of 21 nt, a PBS of 14 nt, and an RTT of 16 nt, they obtained 6.5% of editing. Using this same pegRNA but using PE3 with a sgRNA inducing a nick at position +62, they obtained 10.5% of editing. Because the editing was at position +13 with this peg, the authors tried to use a prime editor that uses another PAM so that the edit site could be closer to the nick. They tried SpCas9n-VQR, which recognizes an NGAN PAM (where the edit site would be at position +1), and SpCas9n-RY, which recognizes an NNN PAM (where the edit site would be at position +3). Even if the edited site was closer to the nick, the percentage of editing did not increase (they each obtained 5.5% of editing). The authors next tried to add a silent mutation in the PAM (G > T at position +6) so that the prime editing event could not happen more than once on an allele. This strategy increased the editing, reaching 7.3% with PE2 and 11% with PE3.
cells-12-00536-t005_Table 5Table 5Prime editing studies on correcting or introducing genetic mutations causing muscle and cardiac diseases.GeneMutationGoal ^1^Prime Editor% of EditingCells or Animal ModelLength (nt)Edit Position from the NickDelivery MethodPrime Editor FormCommentsReferenceSpacerPBSRTT*DMD Exon 52*2-nt AC insertionCPE354ΔEx51 iPSC model201315+4 to +6NucleofectionPlasmidsgRNA inducing a nick at position +52.Chemello 2021 [44]*DMD Exon 59*c.8713C > TIPE26.5HEK293T211416+13Lipo 2000Plasmid
Mbakam 2022 [45,46]PE310.5sgRNA inducing a nick at position +62.PE2-VQR5.51313+1
PE2-RY5.51517+3
PE27.31416+13The PAM was mutated in the RTT (position +6 G > T).PE311The PAM was mutated in the RTT (position +6 G > T) (it had 36% of edition), using a sgRNA inducing a nick at position +62.PE321A mutation at position +19 has been added, using a sgRNA inducing a nick at position +62.PE22819The PAM was mutated in the RTT (position +6 G > T), using a sgRNA inducing a nick at position +62.PE342The PAM was mutated in the RTT (position +6 G > T), using a sgRNA inducing a nick at position +62.PE358A mutation at position +3 (T > C) has been added, using a sgRNA inducing a nick at position +62.CPE317Myoblast211419+13ElectroporationA mutation at position +9 (T > C) and a mutation in the PAM (position +6 (G > T) have been added, using a sgRNA inducing a nick at position +62.PE5max21*DMD Exon 9
*C > TIPE24HEK293T
1313+3Lipo 2000
*DMD Exon 35*C > TPE26+1
G > TPE32010sgRNA inducing a nick at position +57PE21416
PE338sgRNA inducing a nick at position +57*DMD Exon 20*G > TPE2510+10
*DMD Exon 43*C > T3.510+1
*DMD Exon 55*A > T3.513+3
*DMD Exon 61*C > T61513+5
c.428 G > ACPE38Myoblast201416+4ElectroporationsgRNA targeted at +60 nucleotides from the first nick site29After 4 treatments, sgRNA targeted at +60 nucleotides from the first nick site.*SMN2*9 nt deletionCPE316.15HEK293T
1527
jetPRIMEPlasmid
Zhou 2022 [47]29.17SMA-iPSCNucleofection^1^ The goal is either the introduction (I) of the mutation or the correction (C) of the mutation. Abbreviations: Lipo 2000 = Lipofectamine 2000.


The authors next checked whether modifying nucleotides other than the PAM could influence the efficiency of the correction of the desired mutation (at position +13). By adding a mutation at position +19, they obtained an editing of 21% of the desired mutation (at position +13). Combined with the other scenarios where they mutated nucleotides at other positions, their results confirmed that the modification of other nucleotides near it also influenced the editing at a target site. Unfortunately, the mutation induced at position +19 is not a silent mutation, so it could not be used in treatment. However, the mutation in the PAM at position +6 is a silent mutation, so this one could be used in the development of a treatment. They next tested the inclusion of a total of three mutations in the RTT (the target mutation at +13, the mutation in the PAM at +6, and a third additional mutation (C > T) at position +2). By using this pegRNA with PE2, they obtained 28% of editing for the desired mutation (at +13). By using PE3 with this pegRNA, they obtained 42% of editing. The authors also tested the combination of more mutations in the RTT. They observed that the percentage of editing decreased by having a total of five mutations in the RTT. They also wanted to test if the type of nucleotide modification could change the editing efficiency. They tried changing the C nucleotide at position +13 to either an A, a G, or a T. They also simultaneously tested the change of the T nucleotide at position +3 for a C, a G, or an A. The combination of a C > G at position +13 and a T > C at position +3 led to the best result, with an editing efficiency of 58%.

Mbakam et al. [46] also studied the correction of the c.8713C > T mutation in a human myoblast cell line carrying the c.8713C > T point mutation. They used the PE3 system, with a sgRNA inducing a nick at position +62 and with a pegRNA having a spacer of 21 nt, a PBS of 14 nt, and an RTT of 19 nt containing the correction of the desired mutation (T > C), a silent mutation in the PAM (position +6 (G > T), and another mutation at position +9 (T > C). They obtained 17% of editing to correct the desired mutation (+13). They also tried the same pegRNA but with the PE5max system. The PE5max system allows a transient expression of an engineered mismatch-repair-inhibiting protein [48]. It has been showed that the presence of the DNA mismatch repair (MMR) decreases the efficiency of prime editing [49]. Thus, inhibiting the MMR system enhanced the efficiency of prime editing [50]. By using the PE5max system, Mbakam et al. [46] obtained 21% of editing and observed 42% of DMD protein expression.

## 6. Application of Prime Editing to Neurodegenerative Diseases

Qian 2021 et al. [51] studied prime editing for the Tay–Sachs disease. In the Ashkenazi Jewish population, 80% of the case of this progressive neurodegenerative disorder are caused by a four-bases (TATC) insertion in exon 11 of the HEXA gene [52]. The authors introduced this mutation in rabbit embryos to generate an animal model of this disease. They obtained 15.4% TATC insertion using the PE2 system and 37.5% using PE3 (Table 6). The best pegRNA had a PBS of 12 nt and an RTT of 14 nt. Rabbits tissues were sequenced, and one was determined with 68.17% mutation efficiency. The authors also checked for off-targets but did not detect any.

Tremblay et al. [53] worked on the introduction by prime editing of a protective mutation (A673T) in the *APP* gene. This mutation has been shown to prevent Alzheimer’s Disease by reducing the cleavage of APP by beta-secretase [55]. The authors succeeded in inserting this mutation in HEK293T cells with a rate of 6% using PE2, 9.9% using PE3, and 25% by mutating the PAM and using PE3. The additional sgRNA induced a nick at position +79. They also repeated the last prime editing treatment ten times and obtained 65% of editing. For this pegRNA, the mutation was at position +9, and the spacer, the PBS, and the RTT had a length of 20 nt, 11 nt, and 14 nt, respectively. Next, the authors also introduced in HEK293T cells the V717I mutation, which causes a hereditary Alzheimer’s disease. Using a pegRNA with a spacer of 10 nt, a PBS of 16 nt, and an RTT of 17 nt, they obtained 7.5% of editing with the PE3 system and by mutating the PAM.

## 7. Application of Prime Editing to Cystic Fibrosis

Geurts et al. [56] studied the creation of human model organoids to study cystic fibrosis. To create their mutated organoids, the authors used prime editing to induce the disease-causing mutation. After transfecting plasmids coding for PE3, a pegRNA, and a hygromycin resistance gene, clones resistant to hygromycin were selected. They made one intestinal organoid for the F508del mutation and another with the R785* mutation of the *CFTR* gene, mutations causing cystic fibrosis. Although the most known symptoms are in the respiratory system, patients with cystic fibrosis also suffer from intestinal problems. Indeed, fluid transport in the intestine and the organoid entirely depends on the activity of the CFTR channel. Thus, intestinal organoids it a good model for the study of gene correction of mutations in the *CFTR* gene, as the function of this calcium channel can be evaluated.

The F508del mutation is one of the most common in the CFTR gene that causes cystic fibrosis. It is not possible to correct it by base editing [56]. CRISPR/Cas9-mediated HDR was also tried to repair this mutation. Still, the efficiency was very low [57]. Two weeks after the electroporation of the prime editing components, they detected a clone heterozygous for the F508 deletion with restored channel function in the organoid. They used the R785* mutation correction experiment to test for genome-wide off-target effects. Except for a few indels in the region where the additional sgRNA induced a single-strand break, no significant genome-wide differences have been revealed by whole-genome sequencing.

## 8. Application of Prime Editing to Beta-Thalassemia

Zhang et al. [58] published an article about the application of prime editing for a mouse model of the IVS-II-654 mutation that causes beta-thalassemia. Using the PE3 system and a silent mutation in the PAM, the authors successfully corrected the intended mutation in four mice, leading to an efficiency of 14.29%. However, they also generated by-product edits in 32.14% of the mice (nine mice). Fortunately, their off-target analysis showed that prime editing was probably safe for humans. The authors also showed that beta-genotype mice treated with prime editing no longer have hematological symptoms.

## 9. Application of Prime Editing to X-Linked Severe Combined Immunodeficiency

Hou et al. [59] worked on a primary immunodeficiency named X-linked severe combined immunodeficiency (X-SCID). This disease is caused by mutations in the interleukin-2 receptor gamma (*IL2RG*) gene. They first used prime editing to generate an in vitro model of the disease by introducing the c. 458T > C mutation in the *IL2RG* gene in K-162 cells and in healthy donor T cells. They used the PE2 system and electroporated PE2 mRNA and pegRNA. In K-562 cells, they reached 31% of editing. In healthy donor T cells, they reached 26% of editing. They also used prime editing to correct this mutation in patient T cells that carry the mutation with revertant somatic mosaicism. Those patients have somatic revertant mosaicism, leading to an atypical X-SCID with mild clinical symptoms. However, the frequency of the wild-type sequence in the mosaic T cells of patients did not increase after the treatment. This result could be explained by somatic reversion and limited in vitro proliferation of mutant cells. However, the authors hypothesized that prime editing could correct this mutation in patient cells more effectively without revertant mutations.

## 10. Application of Prime Editing to Cancer

Some genes, such as *TP53*, *APC* and *KRAS*, have a predisposition to cause cancer when mutated. It is therefore interesting to study if their correction is possible by prime editing (Table 7).

Abuhamad et al. [60] attempted to edit the *TP53* gene. They used prime editing with PE3 to correct the c.580 C > T, p.L194F mutation in the T47D luminal breast cancer cell line [60]. They used a pegRNA with a 21 nt spacer, a 15 nt PBS, and a 28 nt RTT. Moreover, the editing was conducted at 26 nt downstream of the cut site. With Illumina high-throughput sequencing, they detected only 0.043% editing. The authors also tried introducing the mutation into HEK293T cells. They again obtained meager results, with 0.2% of editing. It should be noted that in their experiments, they also had as a positive control the HEK3 region proposed by the inventor of prime editing technology, i.e., Anzalone et al. [20]. They used the same pegRNA and sgRNA as these authors. This pegRNA changed a C for a G at position 26 downstream of the cut site, thus at the same distance as the peg for the L194F mutation. However, Abuhamad et al. [60] obtained 0.43% editing in the T47D line and 1.3% in HEK293T, which is a meager percentage. Considering that Anzalone et al. [20] obtained a rate of about 35% with this pegRNA in HEK293T cells, questions can be raised about the efficiency of the protocol used by Abuhamad et al. [60]. The extraction of genomic DNA was conducted seven days after transfection. In culture, the percentage of editing decreases with cell expansion as non-transfected cells proliferate better. Their editing rate might have been higher if they had extracted the DNA 72 h after electroporation, as Anzalone et al. [20] and several other authors [25,38] did. Moreover, they isolated only ten cell clones, which is an insufficient number to be conclusive. Moreover, their transfection rate in the T47D line was very low. Optimizing the transfection technique or protocol could led to better results.

Jang et al. [61] worked on correcting *KRAS* mutations by prime editing. A mutation in this gene is the leading cause of many cancers. They designed a unique pegRNA that can correct 12 different *KRAS* mutations, six accounting for 94% of all known *KRAS* mutations. First, to verify whether a unique pegRNA could correct the 12 indexed mutations in the 12th and 13th amino acids of the KRAS gene, the authors established HEK293T/17 library cell lines containing the 12 selected *KRAS* mutations. They then compared the efficacy of the different pegRNAs. Several versions of pegRNA having different PBS and RTT lengths were tested. They found that a pegRNA with a PBS of 13 nt and an RTT of 16 nt was the best. They then compared the effectiveness of PE3 compared to PE2, and it was found that PE3 was better. The authors also tested the efficiency of two versions of engineered pegRNA (epegRNA): the tmpknot RNA motif and the tevopreQ1 RNA motif. Those epegRNA contains a pseudoknot at the end of the 3′ region. These motifs prevent the degradation of the PBS and the RTT sequences of the pegRNA from the cell’s exonucleases. The epegRNAs produced better editing than the classical pegRNA (Table 6).
cells-12-00536-t007_Table 7Table 7Prime editing studies on correcting or introducing genetic mutations predisposing to cancer.GeneMutationGoal ^1^Prime Editor% of EditingCells or Animal ModelLength (nt)Edit Position from the NickDelivery MethodPrime Editor FormCommentsReferenceSpacerPBSRTT*TP53*L194Fc.580 C > TCPE30.043T47D cells211528+26ElectroporationPlasmid
Abuhamad 2022 [60]0.2HEK293T*KRAS*G13Vc.38G > TCPE225HEK293T/17201316
Lipo 2000 or Lipo 3000Plasmid
Jang 2022 [61]PE345.7G12Dc.35G > A46.4epegRNA containing tmpknot RNA motifG12Vc.35G > T44.6
G13Dc.38G > A38.2epegRNA containing tmpknot RNA motifG12Cc.34G > T54.2epegRNA containing tevopreQ1 RNA motifG12Ac.35G > C41.9G12Sc.34G > A45.6G12Rc.34G > C48.3G13Cc.37G > T49.6G13Sc.37G > A49.8epegRNA containing tmpknot RNA motifG13Rc.37G > C50.2epegRNA containing tevopreQ1 RNA motifG13Ac.38G > C45.4G13Vc.38G > T49.1G12Dc.35G > A25G12Vc.35G > T24epegRNA containing tmpknot RNA motifG13Cc.37G > T32G13Dc.38G > A32epegRNA containing tevopreQ1 RNA motifG12Vc.35G > TPE5max11.4CFPAC-1 cellsG12Dc.35G > APE32.7ASPC-1 cellsPE5max18.7krasG12Vc.35G > TIPE314Zebrafish embryos191116+6MicroinjectionRNPsgRNA cut at position +55; pegRNA:PE protein ratio 4:1; the molar ratio of pegRNA/ngRNA in PE3 was 10:1.Petri 2022 [42]^1^ The goal is either the introduction (I) of the mutation or the correction (C) of the mutation. Abbreviations: Lipo 2000 = Lipofectamine 2000; Lipo 3000 = Lipofectamine 3000.


To correct endogenous *KRAS* mutations, they first produced by prime editing HEK293T/17 cell lines containing the specific mutations in the 12th and 13th amino acids. Each clone was heterogenous for the mutation, with two copies of the wild-type KRAS sequence and one copy of the mutated KRAS sequence (chromosome 12, where is located the *KRAS* gene, is triploid in the HEK293T/17 cell line). The authors next corrected the endogenous mutations using the unique epegRNA. For example, for the G12D and G13D mutations, the combination of PE3 and epegRNA containing tevopreQ1 RNA motif was the best, producing 25% and 32% editing, respectively. For the G12V and G13C mutations, the combination of PE3 and epegRNA containing tmpknot RNA motif was the best, yielding 24% and 32% editing, respectively. The authors next tested their prime editing in two pancreatic cancer cell lines, CFPAC-1 and ASPC-1. CFPAC-1 cells had four copies of *KRAS*-bearing alleles, one containing the G12V mutation, and ASPC-1 cells had two copies of *KRAS* G12D mutation-bearing alleles. They first tested PE3 with the unique epegRNA containing tevopreQ1 RNA motif. They obtained a correction activity of only 2.7% in ASPC-1 cells. To improve editing, they then changed the prime editor to use PE5max. This improved their results, with the percentage of editing rising to 11.4% in CFPAC-1 cells and 18.7% in ASPC-1 cells.

Petri et al. [42] studied the introduction by prime editing of the G12V mutation in the *KRAS* gene. This mutation can lead to cancer and cannot be introduced using the base editing system. In this study, the PE3 strategy was used, and prime editing components were delivered in the RNP form. They used a 4:1 pegRNA:PE protein ratio and a 10:1 molar ratio of pegRNA/sgRNA. They used a peRNAg having a spacer of 19 nt, a PBS of 11 nt, and an RTT of 16 nt. The edit site was at position +6, and the additional sgRNA was at the +55 position. With this system, they obtained 14% editing. However, Petri et al. [42] observed significant unintended indels. The observed deletions and potential pegRNA scaffold incorporations in some embryos.

Geurts et al. [56] studied the creation of human model organoids to study cancer. To create their mutated organoids, the authors used prime editing to induce the disease-causing mutation. After transfecting PE3 and pegRNA, clones are manually selected through additional transfection of a plasmid encoding a hygromycin resistance gene. They created several intestinal and hepatocyte organoids with a mutation in the TP53 gene (R175H (c.524 G > A), R249S (c.747 G > T) or C176F (c.527 G > T), for example). Colon organoids were also created with a mutation in the APC gene. For the cancer models, they tested a total of 10 mutations (eight in the TP53 gene and two in the APC gene). Of the 10, the authors could only create organoids for four of these mutations. They also obtained a strange result when attempting to develop the organoid with the APC gene R1450* (c.4348C > T) mutation. Instead of introducing the mutation, sequencing revealed a homozygous duplication of the 37 nucleotides directly upstream of the cut site. This result demonstrates that prime editing can sometimes introduce unwanted changes in DNA.

## 11. Discussion

PE3 definitively improved the efficiency of prime editing compared to PE2. Editing with PE3 can be 5.2 times more efficient than when using only PE2. Even if there are fears about PE3 leading to more off-target mutations, some studies [28,30,37,51] showed no detectable off-target and proved that PE3 was safe. However, other studies showed some off-target mutations [42,56,58]. It is therefore essential to check for off-target mutations, as their frequency seems to depend on the targeted sequence and the pegRNA used. It would be interesting to look at the pattern of these off-target mutations to improve the safety of the prime editing system. Studies also proved that this technology is still more precise than the classical CRISPR/Cas9. Anzalone et al. [20] suggested that the need for three hybridization steps in prime editing explains this high precision. The first hybridization is between the spacer and the target DNA for Cas9 binding. The second is between the PBS and the target DNA, this interaction being essential for initiating the reverse transcription. The third one is during flap resolution, when the RT product binds to the target DNA.

Most studies used a 20 nt spacer sequence, and some used lengths of 19, 21, or 22 nt. The size of the PBS sequence is also critical for efficient prime editing. The optimal pegRNA in most studies had a PBS of 13 nt, which corresponds with the indications of Anzalone et al. [20], who recommended starting the optimization of the pegRNA with a PBS length of 13 nt. Some studies also had better results using a PBS of 9, 10, 11, 12, 14, 15, or 16 nt, which shows the importance of testing a range of PBS-RTT lengths for each mutation to be corrected. The length of the RTT seems to be more flexible, as in many cases, pegRNAs with a given spacer and PBS length but with different RTT lengths perform equally. For PE3, the efficiency of prime editing was more improved when the additional sgRNA was downstream of the cut site.

Mutating the PAM prevents a second prime editing event from happening on an already edited allele. By avoiding the binding of the Cas9 on the DNA, it prevents the loss of the correction. Mutating the PAM has been demonstrated by many groups [24,45,46,53,58] to improve prime editing efficiency. Our group [46] also demonstrated that introducing additional mutations (other than a mutation in the PAM) in the peg can increase prime editing efficiency. We are the only group who has tried this strategy for the correction or the introduction of a human pathogenic mutation. This last approach allows to significantly increase the editing efficiency and to edit sequences that were not thought possible [46,62]. We are also the only group [45,53] which tried repeated prime editing treatments, and this strategy showed a drastic improvement in editing. Those two strategies should therefore be used to optimize the prime editing efficiency. The utilization of epegRNA (containing tevopreQ1 or tmpknot RNA motif) also increased prime editing efficiency [61,62]. Their pseudoknot avoids the degradation by exonucleases of the 3′ region of the pegRNA [63].

Different versions of the prime editor have been used to improve editing efficiency. The use of PE5max increased the efficiency of prime editing by nearly seven-fold compared to PE3 in the Jang et al. [61] study. However, Mbakam et al. [46] also tested PE5max, which only increased their editing by 1.2 times compared to PE3. PE* also showed a considerable increase in editing efficiency. PE2* and PE3* have nuclear localization signal optimization, which improved the efficacy of prime editing up to 3.36 times in vitro and 3.19 times in vivo. Böck et al. [28] used intein-split PE2∆RnH and compared it to the unsplit version. For the dnmt1 gene, they found that the unsplit version was 3.88 times more efficient compared to intein-split PE2∆RnH. For the Pah^enu2^ sequence, the unsplit version successfully corrected the mutation, while the intein-split PE2∆RnH did not show any significant correction. This led to the conclusion that another delivery method than split-PE in AAV should be prioritized.

For the in vivo delivery method, the only ones tested to date are microinjections (in embryos), hydrodynamic tail-vein injections, subretinal injections of trans-splicing AAV2 (for eye targeting), AAV8, and AdV. Delivery by AAV requires the utilization of split-PE, which is not an option to prioritize as it significantly decreases the efficiency of prime editing. AdV should also be avoided in vivo because it triggers an important immune reaction. Injections can only be made for organs that allow non-drastic local delivery, such as the eyes, which can be targeted by subretinal injections [64]. Developing a new delivery method for prime editing is needed for an eventual clinical translation. Some technologies that have already been used to deliver base editing or CRISPR/Cas9, such as virus-like particles [65] or lipid nanoparticles [32], would be interesting to optimize for the prime editing delivery.

Ex vivo treatments were also very effective because the prime editing was administered to cells in culture and not in animals. Thus, transplantation made it possible to return genetically corrected cells to animals. For example, it is very effective in the case of the liver because as the corrected cells have a better proliferation potential, they can repopulate the liver and make it functional again. Thus, targeting an organ where the transplanted cells can take over the diseased cells is more advantageous.

Böck et al. [28] showed that prime editing was more efficient in mouse neonates’ livers than in adults’ livers. This interesting trend should be tested with other targets to better understand this phenomenon. The type of cells targeted by prime editing also impacts the efficiency. For example, Anzalone et al. [20] noticed that prime editing was very efficient in HEK293T cells but less so in other cell lines, such as K562 and HeLa. One hypothesis is that HEK293T cells would be partially defective in their MMR system [48,66], which would favor the conservation of the edit performed by prime editing.

In addition, the form in which the components of prime editing are delivered can significantly impact editing efficiency. For example, Li et al. [33] used prime editing to introduce substitutions in the *SNCA* gene hPSCs. They compared the efficiency when using PE2 in the plasmid, mRNA, and RNP forms and obtained about 5%, 26.7%, and 1% editing efficiency, respectively. Almost all the studies listed in this review used the plasmid form. However, Petri et al. [42] injected the prime editing components as RNPs but did not attempt to compare with plasmid or mRNA delivery. Therefore, no conclusion can be made with this experiment. Hou et al. [59] and Zhang et al. [58] conducted their experiments on prime editing using the mRNA form but did not test their constructions in the plasmid or RNP form. In his thesis, Lung [27] compared the efficiency between the use of DNA or RNA, but the results were not significantly different. Thus, it would be very interesting to compare the efficiency of prime editing in the plasmid, mRNA, and RNP form in the same experiment for the correction of other mutations.

As demonstrated by the studies analyzed in this review, prime editing shows excellent potential for treating inherited diseases. As it is a very recent technology, adjustments are still needed to improve its effectiveness and safety for an eventual clinical application. For example, the prime editing system should be improved to avoid indels, such as when unwanted parts of the pegRNA are incorporated into the DNA. Genome-wide off-targets studies must also be undergone to prove the safety of the prime editing. The potential of prime editing is only at its beginning, as there are still so many genetic diseases for which prime editing therapy could be studied.

## Figures and Tables

**Figure 1 cells-12-00536-f001:**
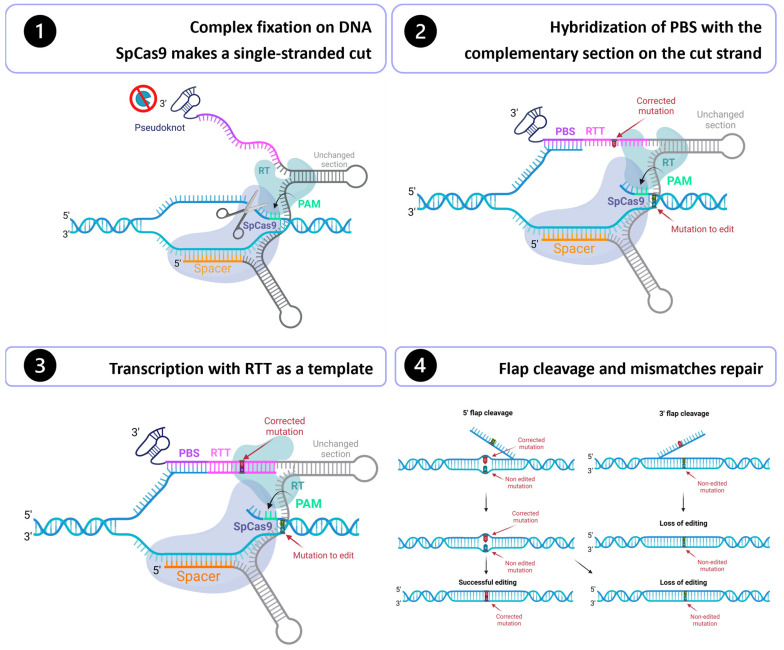
Prime editing mechanism. Step 1: Guided by the spacer sequence in the pegRNA, the Cas9 fused with a reverse transcriptase (RT) binds to the DNA at the desired place in the genome. The Cas9 recognizes a PAM and induce a single-stranded nick 3 nt upstream. Step 2: The PBS hybridizes to its complementary sequence on the cut strand. Step 3: The RT will use the RTT as a template to transcribe the cut strand. Step 4: Mismatches will be repaired by a 5′ flap or a 3′ flap.

**Table 2 cells-12-00536-t002:** Prime editing studies on correcting or introducing mutations causing liver diseases.

Disease	Gene	Mutation		Goal ^1^	Prime Editor	% of Editing	Cells orAnimal Models	Length (nt)			Edit Position from the Nick	Delivery Method	Prime Editor Form	Comments	Reference
								Spacer	PBS	RTT					
Liver cancer	*CTNNB1*	6 nt deletion		I	PE3	30	Liver organoid	20	12	17	+1	Electroporation	Plasmid		Schene 2020 [24]
Bile salt export pump deficiency	*ABCB11*	D482G	A > G			20		20			+7			The PAM is also mutated (+5 G > A silent mutation)	
DGAT1-deficiency	*DGAT1*	S210del	Del CCT	C		21	Patient-derived intestinal cells	20							
Bile salt export pump deficiency	*ABCB11*	R1153H	G > A			0	Patient-derived liver organoids								
Alpha-1-antitrypsin deficiency	*SERPINA1*	E342K	G > A	I	PE2	1.9	HEK293T cells	20	13	27		Lipo 2000	Plasmid		Liu 2021 [25]
					PE3	9.9									
					PE2*	6.4									
					PE3*	15.8									
				C	PE2	2.1	PiZ mice					Hydrodynamic TVI			
					PE2*	6.7									
					PE3	3.1						AAV8			
					PE3	0.83	hPSCs	20	9	13	+3	Electroporation	Plasmid		Habib 2022 [26]
					PE2-NGA	2.0–3.0	HEK293T cells	20	13	20		Lipo 2000	Plasmid		Lung 2021 [27]
					PE3-NGA	3.0–5.0									
					PE2-NGA	1.99	Human primary fibroblasts								
Liver disease	dnmt1	G > C		I	Intein-split PE2∆RnH	15	C57BL/6J pups	21				AAV8	Plasmid		Böck 2022 [28]
					PE2∆RnH	35.9	C57BL/6J adult mice					AdV			
						58.2	C57BL/6J pups								
Phenylketonuria	Pah^enu2^	F263S	T > C	C	Intein-split PE2∆RnH	<1%	Pah^enu2^ mice	20	13	19		AAV8	Plasmid		
					PE2∆RnH	2.0	Adult Pah^enu2^ mice					AdV			
						6.9	Neonates Pah^enu2^ mice								
					PE3∆RnH	11.1									
					PE3	19.6	HEK293T cells			16		Lipo 2000			
					PE3	19.7				19					
Tyrosinemia type 1	fah			C	PEDAR	0.76	FahΔExon5 mice					Hydrodynamic injection	Plasmid		Jiang 2022 [29]
		G > A			PE3	2.3	HT1-mCdHs	20	11	15		Electroporation	Plasmid	sgRNA of PE3 nick in position -4	Kim 2021 [30]
						34.3	HT1 mice					Transplantation			
		c.706G > A			PE3	61	Fah^mut/mut^ mice	20			+10	Hydrodynamic TVI	Plasmid		Jang 2022 [31]
					PE2	33									
	*FAH*					18.7	HEK293T cells					Lentiviral vector			

^1^ The goal is either the introduction (I) of the mutation or the correction (C) of the mutation. Abbreviations: AAV8 = Adeno-Associated Virus Serotype 8; AdV = human adenoviral vector 5; hPSCs = human pluripotent stem cells; Lipo 2000 = Lipofectamine 2000; TVI = tail vein injection.

**Table 3 cells-12-00536-t003:** Prime editing studies on correcting or introducing genetic mutations causing eye diseases.

Disease	Gene	Mutation		Goal ^1^	Prime Editor	% of Editing	Cells or Animal Model	Length (nt)			Delivery Method	Prime Editor Form	Comments	Reference
								Spacer	PBS	RTT				
Leber congenital amaurosis	*RPE65*	R44X	C > T	C	PE2	6.4	rd12 mice	19	9	14	Trans-splicing AAV2 subretinal injection			Jang 2022 [31]
						14	HEK293T cells				Lentiviral vector			
Cataracts	crygc	G-del		I	PE3	80	N2a cells	20	13	10	EZ Trans	Plasmid	Additional sgRNA nicked at position -25	Lin 2022 [37]
						13.8 to 100	B6D2F1 mice				Injection			
				C		33.3	N2a mice				EZ Trans			
X-linked retinitis pigmentosa	*RPGR ORF15*	c. 2234_2237del		C	ePE	12.05	HEK293-mRO cells	20	14	14	TurboFect	Plasmid		Lv 2022 [38]

^1^ The goal is either the introduction (I) of the mutation or the correction (C) of the mutation. Abbreviations: AAV2 = Adeno-Associated Virus Serotype 2; N2a = mouse neuro-2a cells.

**Table 4 cells-12-00536-t004:** Prime editing studies on correcting or introducing genetic mutations causing skin diseases.

Disease	Gene	Mutation		Goal ^1^	Prime Editor	% of Editing	Cells or Animal Model	Length (nt)			Edit Position from the Nick	Delivery Method	Prime Editor Form	Comments	Reference
								Spacer	PBS	RTT					
Recessive dystrophic epidermolysis bullosa	*COL7A1*	c.3631C > T		C	PE3	10.5	Patients-derived fibroblasts and male athymic nude mice	22	13	14	+10	Electroporation	Plasmid		Hong 2021 [41]
		c.2005C > T				5.2					+12				
Oculocutaneous albinism	tyr	P302L	C > T	C	PE3	8	Zebrafish embryo	20	10	15	+3	Microinjection	RNP	sgRNA cut at position -83; pegRNA:PE protein ratio 4:1; the molar ratio of pegRNA/ngRNA in PE3 was 10:1.	Petri 202 [42]

^1^ The goal is either the introduction (I) of the mutation or the correction (C) of the mutation.

**Table 6 cells-12-00536-t006:** Prime editing studies on correcting or introducing genetic mutations causing neurodegenerative diseases.

Disease	Gene	Mutation		Goal ^1^	Prime Editor	% of Editing	Cells or Animal Model	Length (nt)				Edit Position from the Nick	Delivery method	Prime Editor Form	Comments	Reference
								Spacer	PBS		RTT					
Tay–Sachs disease	*HEXA*	4 nt insertion: TATC		I	PE3	37.5	Rabbit embryos		12		14					Qian 2021 [51]
Alzheimer’s Disease	*APP*	A673T	G > A	I	PE2	6	HEK293T	20	11	14		+9	Lipo 2000	Plasmid		Tremblay 2022 [53,54]
					PE3	9.9									sgRNA cut in position +79	
						25									The PAM is mutated; sgRNA cut in position +79	
						65									Treatment has been repeated 10 times; the PAM is mutated; sgRNA cut in position +79	
		V717I				7.5	HEK293T	20	16		17				The PAM is mutated; sgRNA cut in position +48	

^1^ The goal is either the introduction (I) of the mutation or the correction (C) of the mutation. Abbreviations: Lipo 2000 = Lipofectamine 2000.

## Data Availability

Not applicable.

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
