# Peer review of "Prime Editing for Human Gene Therapy: Where Are We Now?"

_cells, 2023, doi:10.3390/cells12040536_

Round 1

Reviewer 1 Report

Authors have done a great job in writing this review article. I have some comments though:

Minor comments: 

  1. Lines 56, 59, 65, and in the rest of the text: Capitalization of 'P' in 'Prime editing and ‘B’ in base editing is not required unless these are the first letters of the sentence. 
  2. Lines 60-61: Authors mention, “The Cas9 and RT will form the Prime editor (PE).”. This sentence should be in the present tense, instead of the future because it is a known fact that the fusion of Cas9 and RT forms the prime editor. 
  3. Lines 77-81: Please clarify or reword the sentence, “As for nomenclature, the reference system is based on the initial Cas9 cleavage site. Thus, nucleotides upstream of this site will have a positive position (e.g., +5 means five nucleotides after the cut site towards 3'), and nucleotides downstream of the cut site will have a negative position (e.g., -5 means five nucleotides before the cut site towards 5').”. The word “nomenclature” refers to the nomenclature of what? It seems the words downstream and upstream may have switched with each other. Upstream is generally 5’ of the reference position and is marked with negative numbering, while downstream is towards 3’ of the reference position and marked with positive numbering. 
  4. Lines 83-85 “It can also make progress studies and research on diseases, as Prime editing can generate cell lines or animal models for a specific mutation.”. Please review the sentence structure. 
  5. Lines 113-114, “They studied the generation by Prime editing of disease model organoids and the correction of patient-derived disease models.”. The sentence needs revision. 
  6. Lines 178-201: The two paragraphs can be combined into a single paragraph and it might be better to describe the modifications made in the respective articles in the chronological order of their publication. The description seems to be overly biased towards the master’s thesis. Since this reference is not a peer-reviewed article, scientific scrutiny of this work is still pending. 
  7. Line 634:639: It might be better to refer to your own group as 'our group', rather than referring to it in the third person as 'Tremblay's group', since it seems that the articles being referenced in these lines are from your lab.  

Major comments: 

  1. Lines 36-37: Authors mention, “The cell will then repair its DNA by non-homologous end joining or homology-directed repair (HDR).” However, apart from the NHEJ and HDR, there is another repair mechanism called MMEJ (Microhomology-mediated end joining) as well. Please refer to below references: 
    • Sakuma, T., Nakade, S., Sakane, Y. et al. MMEJ-assisted gene knock-in using TALENs and CRISPR-Cas9 with the PITCh systems. Nat Protoc 11, 118–133 (2016). https://doi.org/10.1038/nprot.2015.140 
    • Sfeir A, Symington LS. Microhomology-Mediated End Joining: A Back-up Survival Mechanism or Dedicated Pathway? Trends Biochem Sci. 2015 Nov;40(11):701-714. doi: 10.1016/j.tibs.2015.08.006. Epub 2015 Oct 1. PMID: 26439531; PMCID: PMC4638128. 
    • Zhang C, Meng X, Wei X, Lu L. Highly efficient CRISPR mutagenesis by microhomology-mediated end joining in Aspergillus fumigatus. Fungal Genet Biol. 2016 Jan;86:47-57. doi: 10.1016/j.fgb.2015.12.007. Epub 2015 Dec 14. PMID: 26701308. 
  2. Lines 42-43: The authors have mentioned, “This system exists in two versions: cytosine base editors (CBEs) [6] or adenine base editors (ABEs)”. But In addition to CBEs and ABEs, there are C to G base editors as well. Please refer to the below-mentioned references: 
    • Kurt IC, Zhou R, Iyer S, Garcia SP, Miller BR, Langner LM, Grünewald J, Joung JK. CRISPR C-to-G base editors for inducing targeted DNA transversions in human cells. Nat Biotechnol. 2021 Jan;39(1):41-46. doi: 10.1038/s41587-020-0609-x. Epub 2020 Jul 20. PMID: 32690971; PMCID: PMC7854778. 
    • Cao T, Liu S, Qiu Y, Gao M, Wu J, Wu G, Liang P, Huang J. Generation of C-to-G transversion in mouse embryos via CG editors. Transgenic Res. 2022 Oct;31(4-5):445-455. doi: 10.1007/s11248-022-00313-x. Epub 2022 Jun 15. PMID: 35704130. 
  3. Lines 50-51: Authors mention, “However, this system cannot perform the eight transversion mutations (C>A, C>G, G>C, G>T, A>C, A>T, T>A and T>G),…”. This statement is incorrect. Please, refer to comment no. 2 above for lines 42-43 about C to G base editors. 
  4. Lines 61-62: Please elaborate on "an unchanging part that fits with the Cas9 and the RT" in line 62.  
  5. Lines 120-121: Gene ABCB11 does not cause Wilson disease. This error is repeated in table 1 as well. Moreover, the capitalization of ‘D’ in ‘Disease’ is not needed. 
  6. Lines 633-637:  Authors mention, "Mutating the PAM has been demonstrated by many groups [11,32,33,40,45] to improve Prime editing efficiency. Tremblay’s group [33] is the only one that introduced additional mutations (other than a mutation in the PAM) in the peg for the correction or the introduction of a human pathogenic mutation. This last approach allows to significantly increase the editing efficiency and to edit sequences that were not thought possible.". Other authors have also done modifications at the 3' end of the pegRNA. For example, please see figure 3b in the below reference: Liu, P., Liang, SQ., Zheng, C. et al. Improved prime editors enable pathogenic allele correction and cancer modeling in adult mice. Nat Commun 12, 2121 (2021). https://doi.org/10.1038/s41467-021-22295-w 

Author Response

See file.

Reviewer 2 Report

The manuscript Prime Editing for Human Gene Therapy: Where Are We Now? by Godbout & Tremblay provides an in-depth look at the current advancements in Prime editing technology with recent publications and work in progress. This overview facilitates the reader's understanding of the working mechanism of the technologies; however, it lacks the complications, adverse related events and necessary improvement needed to reach translational potential.

The manuscript reads with fluidity, I like the tables with the summary of the findings. I am missing a recap or summary for each group of diseases, a conclusion of the current work and next steps. Maybe, to try to describe the apers not as a list one after the other but finding points in common between the works and differences.

Major comment

- A table summarizing the advantages and disadvantages of traditional Crispr, Base editing and prime editing will benefit the manuscript

Minor comments.

- Introduction about gene therapy is make too blunt, and rush into the Cas9 system. a better understanding of the modalities of gene therapy and how CRISPR is part of it will give more fluidity to the manuscript and better understanding to the reader. 

-Line 36 is missing the abbreviation (NHEJ)

Author Response

See file.

Reviewer 3 Report

Generally I found the review to be interesting and informative providing a comprehensive overview of the PE studies performed to date - specifically as applied too human diseases or models thereof.

I recommend a systematic review of the English syntax though generally it is very well written. (e.g.L62: unchanging part = common region; L83-84: 'make'?, L86: on = using, L170: nick, L180-183: is this a repeat from earlier?,  L609:  'can reach up to 5 times the one obtained' - poor grammar)

I would avoid the copy and paste of text from the main body into Fig1 legend (repeats)

L200: what is the function of TREX2

L465: 'Rabbits were sequenced' - the entire rabbit? Or cells/tissues from....

Could the authors comment on the potential application of PE in mitochondrial DNA editing?

Author Response

See file.
